# “*COVID Is Another Layer of Problematic Things*”: Change, Vulnerability, and COVID-19 among University Students

**DOI:** 10.3390/ijerph192315947

**Published:** 2022-11-30

**Authors:** Ifeolu David, Enid Schatz, Tyler W. Myroniuk, Michelle Teti

**Affiliations:** 1Department of Health and Rehabilitation Sciences, School of Health Professions, University of Missouri, Columbia, MO 65211, USA; 2Department of Public Health, School of Health Professions, University of Missouri, Columbia, MO 65211, USA

**Keywords:** academic, college students, COVID-19, experiences, mental health, qualitative

## Abstract

The COVID-19 pandemic not only had detrimental effects on physical health but also had adverse effects on college students’ mental health. This paper begins to fill a gap in knowledge related to the contextual factors that impacted college students’ mental health during COVID. Using in-depth interviews with a diverse sample of 33 college students at a Midwestern university, during Spring 2021, we highlight the pandemic’s role in shaping college students’ mental health and their outlook of the future. Thematic analysis revealed student reports of mental health decline during the pandemic attributed to campus closures and social distancing policies implemented by the institution to reduce the spread of COVID-19. Students shared that the pandemic created uncertainties about their future opportunities for education, career fulfillment, and employment. However, the interviews also suggested a general sense of adaptation to the pandemic’s impact which was students achieved via a combination of active and passive coping strategies. Expanding institution-based mental health services to include a variety of modalities and off-line toolkits for students can help students cope with mental health challenges, whether in ‘normal times’ or during national crises. Future research should focus on identifying strategies for promoting mental wellness among college students and exploring post-pandemic mental health wellbeing.

## 1. Introduction

Since March 2020, the COVID-19 pandemic has impacted all aspects of life and society globally [1]. In addition to having a direct negative impact on physical health, the pandemic and associated lockdowns and restrictions also negatively affected mental health due to fear, isolation, and anxiety [2]. Insights from social stress theory suggest that individuals are susceptible to adverse mental health outcomes when their social circumstances are unfavorable [3]. In the context of the COVID-19 pandemic, mental health challenges could include irritability, anxiety, depression, and stress [4]. Initial interventions such as social distancing, shelter-in-place, lockdowns, and self-isolation, aimed at mitigating the physical health impacts of the pandemic came with mental health side effects [5,6,7,8]. These included a high prevalence of anxiety, depression, and stress [6,9]; a group notably impacted by these pandemic mental health challenges were college students.

Even before the onset of the pandemic, college students experienced high levels of depression, anxiety, and stress [10,11,12,13]. According to the American college health association-National college health assessment III in 2019, more than one-third of students reported at least one mental health symptom [14]. Research shows that these psychological challenges are associated with changes in residence, lifestyle, peer influence, learning, and work obligations [15]. Despite the high prevalence of mental health issues, college students are less likely to utilize mental health services than the general population [16,17].

While college and university closures or move to online learning was a necessary COVID-19 prevention strategy, there were significant unintended impacts on the academic and social lives of college students [1,18]. The transition to online learning was abrupt and left students with little time to adapt to changes in living circumstances, learning, and social interaction. Consequently, many college students who experienced disruptions in their academic routines and the shift to online learning reported poor mental well-being [1,8]. About a quarter of college students in China, and 61% in France reported anxiety as a result of the COVID-19 pandemic [19,20]. Nearly 80% of students in the College of Engineering at San Jose State University in the US, reported moderate to severe levels of stress related to the shelter-in-place mandate [21]. Although some of these experiences are attributable directly to the closure of universities or the sudden shift to online learning, others may be linked to changes in lifestyle and social situations during the pandemic [18]. Risk factors for depression during the pandemic included identifying as female, staying at home, having a history of chronic illness, and having poor and moderate social support [22,23]. Additionally, not living with parents, low family income, and having family members who had COVID-19 were also risk factors for anxiety [24]; students who were not living with their parents had a 3.3 times higher risk of developing anxiety compared to those who were living with their parents [22,24].

Researchers have focused a lot of attention on the COVID-19 pandemic’s effects on mental health, but less is known about how these effects have shaped college students’ experiences or students’ outlook on the future. Much of the published work to date is quantitative, highlighting the prevalence of mental health symptoms. This paper seeks to expand the literature on college students’ mental health during the pandemic using qualitative interview data to highlight students’ experiences of declining mental health, shifting perspectives on the future, and coping. Our study of a diverse sample of 33 college students at a Midwestern University (MWU) offers depth and insight into the perceived reasons for depression and anxiety. Even though most US college campuses have drastically reduced or entirely removed COVID-19 restrictions, it is important to understand how students’ mental health was impacted by the COVID-19 pandemic, in efforts to provide ongoing support and to better prepare for future pandemics.

## 2. Method

### 2.1. Sampling and Data Collection

Our qualitative data originate from students at Midwestern University (MWU) who participated in a survey conducted in the Fall of 2020 as part of a study titled, “Seropositivity and Risk for SARS-CoV-2 and COVID-19” (N = 1446 undergraduate and graduate students). At the end of the survey, students were asked if they were willing to be contacted for follow-up interviews; the majority (77%) consented and thus served as the sampling pool for this study. We purposefully sampled students to ensure that the qualitative sample was diverse in terms of gender, race, and sexual orientation; students were recruited from the sampling pool via email. The final sample included 10 students randomly selected irrespective of personal characteristics (sampled from N = 1155); 13 non-white students (sampled from N = 117); and 10 self-identified LGBTQ+ students (sampled from N = 151) as shown in Appendix A. In order to accurately represent the perspectives of the heterogeneous study population, we interviewed to reach saturation in thoughts and experiences linked to COVID-19 across these demographic categories.

### 2.2. Interviews

At the participants’ request, interviews were conducted by phone or Zoom and lasted around 30 min. Interview questions focused on a variety of elements of students’ life throughout COVID-19, including academics, relationships with friends and family, illness, and mental health (see Appendix A). These questions, together with others regarding participants’ opinions and behavior in relation to COVID-19 and the university’s COVID-19 prevention effort, served as the foundation for this analysis. For analysis, interviews were audio recorded, transcribed verbatim, and assigned pseudonyms.

### 2.3. Analysis

Using thematic analysis, the first three authors reviewed the transcribed data independently to generate a list of themes around mental health experiences. The authors subsequently met to discuss, compare and consolidate ideas—developing major themes to comprise a codebook [25]. Using the codebook, the research team analyzed three transcripts each before meeting again to further clarify the codebook [26]. After agreement on code usage was reached, the final version of the codebook facilitated analysis of the remaining transcripts. As coding occurred, the team met regularly to discuss common themes and unique findings. Although our sample included students who identified in a variety of ways—given the similarities in responses related to mental health across demographic groups, we do not provide separate analyses by group. For this paper, excerpts were exported and categorized into three main themes that went through further analysis for sub-themes. These themes and sub-themes are presented in the findings below.

## 3. Findings

The pandemic affected our participants’ lives in ways that shaped their perspectives on mental health issues. Changes in what was previously normal with family, education, and social routines dampened emotions and resulted in three major categories related to mental health: (1) experiences of mental health decline, (2) changing perspectives and ruminations about the future, and (3) adapting to the pandemic’s impact. The students identified occurrences of anxiety, depression, and stress during the pandemic as they made adjustments to their everyday routines. For some, struggles with mental health were new, while for others, the pandemic exacerbated pre-existing mental health conditions. Concerns about COVID-19 infection and its potential short- and long-term health consequences were widely discussed. Beyond their own social world, participants expressed concerns about the pandemic’s possible implications on social and political strife and how these might affect society as a whole. Through this lens, they discussed worries about future career, economic, and higher education opportunities. In response to dealing with these personal and social impacts, participants reported a variety of response mechanisms that they employed to deal with the pandemic worries while being intentional about maintaining optimal mental health despite the challenging context.

### 3.1. “This Year Has Tested My Mental Health”

The COVID-19 pandemic presented a new set of challenges and stressors for college students. Campus closures and return to family homes meant that social activities were few or absent for most students. Still, students had to keep up with school demands virtually, a feat that required extensive efforts to manage and accommodate academic work in non-academic environments. At the same time, public health recommendations for reducing social interaction to reduce transmission led to students feeling isolated and limited in their college experience. Our participants universally discussed mental health declines and identified specific incidents of stress, anxiety, loneliness, and depression citing the pandemic as having had a significant impact on their mental health.

Students experienced attending to the new social and academic landscape as stressful. Thriving during the pandemic presented unique challenges making academics and everyday activities a constant source of stress. Jeff, a White, LGBTQ student, bemoaned these issues.

Um, I definitely would say that this year has tested my mental health. If I had to compare it to all my other years of college, it would be the worst year. I know there are resources out there available to me, but I feel like everyone just kind of plays it off like, ‘Oh, it’s the pandemic’ so I’m just trying to push through and graduate. I have 5 weeks now of this semester but a big toll on my mental health has been because of my work. And, because my work has been completely altered by COVID. I used to love my job, going there almost every single day and now it’s just stressful. The turnover because of COVID has increased so it’s more job responsibilities on myself.

Even activities that Jeff loved or experienced as joyful prior to the pandemic led to stress and discontent.

Anxiety was raised in relation to everyday life activities, family, and academic work. Natasha who identifies as a White, LGBTQ student shared, “[the pandemic] taught me that I am susceptible to anxiety. I never really felt anxiety like that, kind of like a pervasive anxiety, from COVID. And I’ve had lots of drama and trauma in my life, but it was a different kind of worry and anxiety that wasn’t healthy.” Experiences of anxiety were reported in relation to potential exposure to COVID-19 as the interviewers were occurring at a time when there were many reports of severe illness and death resulting from COVID-19 infection and regular reports of cases on campus.

A number of our participants expressed a strong desire to stay healthy during the pandemic and dreaded the thought of possible infection resulting from exposure to COVID-19. Kelley, a White LGBTQ student, highlighted this sentiment. She said she was aware of her, “mental health, [I have] a lot more anxiety. But [my] physical health is mostly the same. Yeah, definitely increased anxiety from not wanting to get sick. Or if I did get sick, worrying that it was COVID. So that was the biggest mental health thing.” Anxiety for these students influenced how they went about their ‘normal’ life activities. COVID-related anxiety was pervasive and felt uncharacteristic for many of the students who experienced this anxiety.

Social isolation was a key factor in deteriorating mood impacting feelings of loneliness and depression; our participants struggled to maintain optimal mental health in the absence of the social and community experiences associated with higher education. Students connected loneliness to mental health decline. Benjamin a non-White male student said, “At times, I would say that it [my mental health] declined just because I’ve felt lonely.” Jasmine, a White female, LGBTQ student, shared similar experiences, and as the pandemic progressed, her mental health decline intensified. She stated, “it’s now been kind of bumpy because I’m getting to the point where I’m more burnt out on the pandemic, and just sick and tired of spending so much time by myself in my apartment.” This sense of social isolation and loneliness may have facilitated lowered transmission, but the cost was the mental well-being of students like Jasmine and Benjamin. Noah, a non-White male student, shared his experiences, remarking, “And [the pandemic has] led to me like, just like staying in my room, and staying in my room like one place that’s made me sometimes like, in a depressed mood. Um, definitely more prone to depressing episodes.” Suggesting that isolation was not just experienced as episodes of loneliness, but also of feeling depressed.

While some students were able to manage depressive episodes with physical exercise or other positive coping strategies, the proximity to COVID infection and deaths, and well as the multiple waves of the pandemic meant that many students cycled into and out of depressed states and some managed their feelings with less ideal coping strategies. Jasmine a White, LGBTQ student noted that.

In general, and this is something I have personally noticed, is, a lot of people’s, like, substance abuse—well, substance use, I should say, use, not abuse—use has gone up significantly throughout all of this. And I personally can attest, I’m smoking a lot more weed than I should be, probably, and drinking and stuff, because it’s—you know, you’re burnt out, and you’re exhausted. And I’m seeing that trend a lot with, just, people in general. And especially with college students, because you already have that culture where a lot of that is accepted.

For some students managing pandemic challenges led to new experiences of burnout and mental health decline. For a number of our respondents with preexisting mental health conditions, however, the pandemic aggravated their existing symptoms or conditions.

For some, their mental health condition worsened to the point where they felt the need to seek additional professional help. Kalia a White, LGBTQ student emphasized her experience with mental health during the pandemic and she recognizes the role of the pandemic in her remarks “[My] mental health definitely went down. I actually got a therapist a few months ago. I don’t think it was onset from COVID-related things, but COVID is another layer of problematic things, you know?” She did not ascribe COVID as the cause of her mental health condition but did feel it contributed to her need for a therapist’s help.

One respondent experienced a relapse of a mental health condition she had been successfully managing. Rebecca, a White female student, shared her experiences with the pandemic’s impact on her depression, “God, it’s [COVID] changed everything. I mean, you listed quite a long list there. Where should I start? My mental health has taken a big hit. I was diagnosed with depression probably three or four years ago when I was on vacation. I’m actually going to get back into therapy.” Rebecca’s comments highlight how the COVID-19 pandemic had aggravated pre-existing mental health problems.

Social seclusion and distancing that were common in the first year of the pandemic proved particularly difficult for individuals with pre-existing mental health challenges. For Lloyd, a White, LGBTQ student loneliness during the pandemic was a precursor to depression episodes. While he had been dealing with depression prior to the pandemic, his symptoms worsened increasing his medication needs.

Yeah, I would say in that kind of isolation, I have been more depressed. I started seeing a therapist again recently, which I had done prior. And I’m also on SSRIs, so selective serotonin reuptake inhibitors, so I have been put on them in increasingly higher dosages since COVID and I have only started on them in COVID, so last fall is when I started that medication, so a definite decline in mental health following COVID, not that I had great mental health before, but some of my predilections towards anxiety and depression increased during COVID.

As evidenced above, the pandemic took its toll on college students whether they had pre-existing conditions or not, students experience a variety of negative mental health consequences related to fear, isolation, and missing out on a ‘normal’ college experience.

### 3.2. Sensemaking: Changing Perspectives

For many students, the pandemic experience involved more than just mental health symptoms; it also affected how they viewed the world. Respondents noted that the COVID-19 pandemic had uncovered vast differences in personal, religious, and political beliefs leading to worries about their own futures in terms of ways the pandemic could impact their academic goals, career path, and employment opportunities. While some of our participants did not identify these concerns as mental health issues and hence did not discuss them in relation to stress, anxiety, loneliness, or depression, all narratives that mentioned these issues offered insights into the emotional landscapes experienced as the pandemic progressed.

Participants expressed worries about the implications of the COVID-19 outbreak on societal norms and values. Due to conflicting beliefs and perceptions around COVID-19, public emotional flares had not been uncommon. These experiences contributed to emotional dampening and were a source of concern during the pandemic. Martin, a non-White male student, considers the pandemic’s impact on societal values his “biggest worry” and explains how he feels about the situation.

I think my biggest worry is how we’re going to get back to what is normal. I personally feel like people have overreacted, and then there are people who have underreacted to it. So, I think that my biggest issue is that, as a society, we aren’t able to find a common ground or whatever on how to get back to what’s normal.

Jimmy, another non-White male student, explained similar concerns about the strains on normality and interpersonal relations in public spaces.

I think more of not COVID itself but you know you see in a society where people are like extreme on wearing PPE and social distancing. Even here in the clinic, we have had patients get very aggressive. People out there trying to be a little too cautious and therefore they are getting aggressive. You see it at the stores and people call the cops on people. I have not personally seen that. That is my only concern. Eventually one day we have a lot of people out there upset because they are not doing what someone else expects them to do. That wasn’t a big deal before. No one ever yelled at you if you had a runny nose and you weren’t wearing a mask.

Students expressed concern about other tears in the social fabric. Rebecca, a White female student, shared fears for altered family and job situations and she shared her thoughts with the following remarks, “I worry about people who have lost their jobs, people who are struggling to feed their families, people who are on the brink of getting evicted, losing their homes. There are so many things.”

The uncertainties and concerns went beyond what had already occurred to worries about changes in future opportunities for education, career fulfillment, and employment. Our participants believed that the economic downturn associated with the COVID-19 outbreak could have dire consequences on future aspirations which made students feel more stressed during the pandemic.

Janelle, a non-White female student, shared her thoughts about the abrupt disruption in opportunities for students to gather necessary experiences as they prepare for various career trajectories.

But yeah, I guess the biggest worry is worrying about these consequences whether it’s within this college or within the country, the world, whatever, in the future because a lot of students, especially in STEM fields who want to get into labs, people want to try to get experience before they go to med school or whatever they’re doing are stopped because they can’t get into labs. So, I really worry about the future in terms of career goals and stuff like that.

Even though the transition to online learning was a success, students pursuing academic fields that require in-person training missed out on key experiences. Shawn, a non-White, non-binary student went further by citing their experiences as they felt that the COVID-19 disruptions could delay his academic goals.

I’m worried that with COVID, I haven’t [had a chance to do] as much research as I would have previously which might affect my graduate school application. I might—in the past couple of weeks, I’ve been thinking what if I take another year after graduation to do more research instead of going straight to graduate school. But my career goals are still the same. I want to grad school, and I want to do research.

In addition to fears about careers and furthering their education, participants expressed a general feeling of uncertainty about the future. Not knowing what to expect was particularly uncomfortable. Grace, White female student, articulated this discomfort in this way.

I feel like in general, just how it’s going to impact our country and world. Also, jobs I feel like it’s going to impact because I know so many people have already lost their jobs. So, just how it’s going to change that and just change our dynamic in the world, in general, is a big fear of mine.

For our participants, uncertainty, and apprehension about the future intertwined with their experience of the COVID-19 pandemic. While uncertainty about the future may be common at this stage of the life course, in Spring 2021 this uncertainty was connected with negative emotions (fear, worry) that were also linked with negative mental health outcomes [27]. While these impacts were real, students also highlighted their agency in coping with these new burdens.

### 3.3. Adapting to the Pandemic’s Impact

Some of our conversations with participants revealed a sense of adaptation rather than helplessness. Despite the challenges to goal achievement posed by social isolation, individuals enacted practices to address the emotional slump associated with the pandemic. Coping strategies included actively taking control of their lives or more passively engaging in optimism; most students used a combination of active and passive strategies.

#### 3.3.1. Actively Taking Back Control

The pandemic increased the visibility and awareness of mental health issues making students feel that it was more acceptable to discuss mental health and do something about it. Students took agency to manage their mental health in a number of forms: professional care including therapy, courses, and medication, as well as building community online and in-person.

According to some participants, the decision to seek mental health care was motivated by how the pandemic itself had contributed to mental health declines. Some students noted this was their first time seeking formal mental health care. Ana, a non-White female student, shared her experiences with the following remarks.

In terms of anxiety though, I’d say that COVID has made me more willing to seek out help because it’s more normalized, I guess? So, I did take an [anti-] anxiety course through the counseling center, back when I was rooming with the girl who didn’t care about COVID, and that was one of my biggest sources of anxiety. I do have a generalized anxiety disorder, and my workload is a huge contributor, but COVID is also a big part of it. Just struggling to fit in and make friends on campus when I care a lot. So I took that course and then this semester I actually reached out to a psychiatrist and got medicated for anxiety for the first time and that’s been beneficial to me as well.

While Kalia believed that the pandemic exacerbated rather than was the source of her mental health decline, the additional ‘layer’ led her to seek help, “Yeah. Mental health definitely went down. I actually got a therapist a few months ago. I don’t think it was onset from COVID-related things, but COVID is another layer of problematic things, you know?”

Kelley outlined her coping process in this way, “I did take strides to help myself. I went to group therapy and didn’t let myself just sit at the computer all day long. I at least got up and I did something else in the apartment. I think it kind of helped me there not get into a depression.” As mental health therapy services were offered virtually, participants also took advantage of this opportunity. For Janelle, virtual therapy visits were not something she had done before the pandemic but were an integral part of her path to taking back control of her mental health,”… I see a therapist virtually, and I started seeing her in June of 2020. I haven’t stepped foot in the office ever. I’ve only known her for this computer screen. That was something that was interesting because I’m not used to seeing therapists on the computer screens, it’s usually in person.” The availability of both in-person and virtual services was vital for college students. Even though some had not used such services before, each of these students took action to seek care to address what they saw as declining mental health related to the pandemic.

Forming, maintaining, and leaning on social communities was another common coping strategy among our participants. These virtual or in-person communities provided needed social interaction and social support.

In some instances, these communities were achieved virtually via cooperative activities such as video gaming. Lucas, a non-White male student, disclosed his ADHD made social isolation particularly challenging; he found, however, that online video gaming provided him with needed social interaction, but also not having to just sit still.

Hell, yeah. I’d say it’s difficult, to say the least because I’m registered through the disability center. I have moderate to severe ADHD. I feel like it’s a need for me to go out and move. It’s really hard for me to sit down. So, I’ve just found different things like—I don’t know. One of the necessities I made when we first came up here with my brothers was everybody had some type of Playstation so we could talk to each other through the headset. We could play with each other. It was some type of indirect social interaction. I’d make sure I talked to my brothers, I talked to my older sisters, all through kind of that. And it was just a way to keep my hands busy because I really can’t go outside and get my own body energy out of there.

Alex, a non-White male student, reported similar practices, “Another thing is how much more we use virtual sources, for example, now whenever my friends can’t really hang out, like my friend group, we’ll go on Discord, and we’ll all be playing video games together, but separately.”

Other participants opted for in-person social communities including friends with whom they share similar health and behavioral attributes. Students formed social “bubbles” where they interacted with a small designated group of friends. While social “bubbling” was not a perfect COVID-19 prevention practice, was a good risk mitigation strategy that reduced the chance of transmission while allowing for social interaction with a select group to reduce the adverse consequences of loneliness among members. Social “bubbling” also provided some form of accountability towards observing infection prevention practices since members would not want to risk exposing others through negligence. Janelle described her bubble in this way.

It’s just me and like two or three friends. And it’s us together in general. And so, we were bubbling before bubbling was cool and everything. And so, we still keep ourselves safe. And we’re all scientists so we believe and listen to what’s going on. We’re not going to bars and we’re not doing anything like that… My friends are the ones that are keeping me sane. Us, each other, we’re kind of there for each other and keep each other grounded and not have cabin fever or whatever.

Noah described a similar situation, “But uhh, in the beginning, like, the pandemic was like, really scary, and like I wasn’t seeing any of my friends, but I had like this small bubble, and all of my friends have just been hanging out inside that small bubble of like, like, 4 to 7 people. So like, a decent amount, but those are the only people that I’m not social distancing with.” This active risk mitigation strategy allowed students to have social interaction to address mental health needs while limiting COVID exposure.

#### 3.3.2. Optimism as a Form of Coping

As the pandemic progressed, students felt it was increasingly doubtful that things will be returning to “normal” soon. And yet, a sense of optimism and hope was a major source of comfort for many of our participants. Optimism and hope were spurred by news of COVID-19 vaccine approval and availability, but also by a desire to stay positive in a difficult situation. Alex expressed his thoughts on this “I think America is hopefully close to the end of the tunnel here. I’m seeing the light at the end of the tunnel, but I don’t know if that’s accurate or not”.

COVID-19 vaccines represented a major source of hope and optimism about the pandemic’s end for a certain group of students. Since vaccines prevent infection and reduce transmission of infectious diseases, it was not surprising that college students viewed vaccine availability as a major breakthrough in the pandemic. Courtney explained her perspective, “But, I think, overall, there’s still a lot of hope. Especially with the vaccine coming out and that kind of thing. I think—especially with my friends. We’re just counting down the days and just hoping that soon, it’ll all just slowly start going back to normal I think.” Kalia was lucky to have been among the first groups of individuals eligible for the vaccines, “But, I got the first vaccine, and that has definitely brought down some anxiety about it.” For her, receipt of her first vaccine dose played a crucial role in reducing anxiety about possible COVID-19 exposure.

When asked about thoughts on the future and whether they felt worried, rather than focusing on the challenges, some students felt COVID-19 was providing them with lessons and preparing them for the future. Noah shared his belief that the COVID-19 pandemic was preparing the world to combat potentially more fatal health challenges later, “Um, not really. I’m actually pretty optimistic about the future because I think that while COVID-19 was bad, it was honestly like, one of the best-case scenarios for global pandemics. And I think that this would, this will shock people enough to be prepared for a worse epidemic in the future.” He went on to highlight the ways pandemic health and workplace practices have ingrained important lessons.

I think that masks will be more common when people are sick for anything. I am looking forward to that. And I think that there will be less pressure for people to come to work when they have like the flu or something, I think people would be more conscientious about illnesses and contagion. So my outlook on the future is very optimistic, and that might be naive, but I can hope.

Even though little was known about the effectiveness of the vaccine at the time of our study, optimism among our participants allowed students to feel hopeful about better conditions in the future, which may have protected them against some of the negative mental health effects of the pandemic.

## 4. Discussion

The COVID-19 pandemic affected college students’ lives in many ways. Our participants reported experiences of mental health decline during the peak of the pandemic when COVID-19 prevention policies led to campus closure. Additionally, the pandemic experiences of our participants had an impact on their perspectives of the future, creating uncertainties about societal norms and values as well as future opportunities for education, career fulfillment, and employment. Lastly, participant conversations indicate a general sense of adaptation to the pandemic’s impact which was achieved via a combination of active and passive coping strategies.

Identified experiences of anxiety, depression, and stress were discussed in relation to changes in daily routines during the pandemic. For some, mental health decline was a first-time experience and for others, it was an exacerbation of pre-existing mental health challenges. By uncovering connections between the COVID-19 pandemic and mental health outcomes, our findings build upon existing research showing that college students experienced high levels of depression, anxiety, and stress during the COVID-19 pandemic [1,16,21]. Lee et al. (2021) found that 80% of their undergraduate study population at a Kentucky university experienced moderate or severe stress while another 36 to 44% displayed moderate or severe anxiety and depression. Loneliness resulting from limited social interactions was noted by our participants as a precursor to mental health symptoms and hence was thought to have a major role in mental health decline. Studies conducted among university students in France and Canada during the pandemic also revealed that social isolation resulting from COVID-19 prevention practices was strongly associated with declining mental health [18,28].

College students also reported that pandemic-induced concerns for the broader society impacted the ways they were thinking about the future. Students discussed concerns about the job market, social distancing, and the political divisions resulting from conflicting beliefs about the COVID-19 infection. Our participants were therefore concerned that life may not return to what we know as “normal”. As students, they had additional concerns about opportunities for quality educational experiences, career advancement, and employment. This is similar to findings from existing literature among college students in the USA which suggest that concerns about academics, one’s health, job, employment, and uncertainties about the future were frequently reported among college students and were strongly linked with declining mental health [23,29,30].

Coping strategies were an essential part of our participants’ adaptation to changes in lifestyle during the pandemic. Some participants maintained a sense of optimism and reportedly invested time in maintaining close-knit social groups virtually and sometimes in-person while others sought professional help in the form of counseling and therapy. Like our study findings, college students have demonstrated more optimism about the future in spite of high-stress levels experiences. This was evident in Chierichetti & Backer’s (2021) study from Fall 2020 involving students at the San Jose State University, where more than half of the 400 study participants reported being positive about their long-term goals and prospects, which helped them deal with pandemic challenges better [21].

Contrary to existing discourse suggesting that college students are more likely to ignore or be reluctant to identify and accept mental health challenges [31,32], our participants discussed occurrences of anxiety, depression, and stress experienced during the COVID-19 pandemic. Further, most of our participants reporting mental health symptoms showed willingness and sought formal help during this time. This was surprising given that a large-scale study of American college students showed that during the COVID-19 pandemic the majority who reported mental health challenges did not seek available services [16,17]. Additionally, save for one respondent, the majority of our participants’ discussions of coping did not include negative strategies like substance overuse. Coping among college students is predicted to include a combination of positive and negative practices as seen in other literature exploring USA college students during the COVID-19 pandemic [21]. Studies conducted among students attending universities in France and the USA found that students had increased consumption of tobacco and alcohol to cope with stress during the pandemic [17,18]. The lower use of negative coping in this study may be explained by the fact that the majority of participants who reported mental health decline sought professional assistance.

Thus, our findings describe college students’ experiences during the pandemic and highlight important lessons from those experiences. We also recognize that the COVID-19 pandemic contributed to a decline in mental health, even among students who had no prior history of such problems. For college students with pre-existing mental health conditions, the COVID-19 pandemic presented “an extra layer of problems”.

## 5. Limitations

Mental health challenges reported in this study were based on participant self-report rather than clinical evaluation, but we believe that self-reporting was ideal to elicit discussions around mental health experiences and actions of college students during the pandemic. Additionally, the sampling for our study was intended to highlight the experiences of a diverse set of college students but did not aim to be representative; thus, while we met saturation in our findings, and the themes represent findings across groups, caution must be used in generalizing these results to the student body at our own or other academic institutions. Our findings, however, suggest that college students at large predominantly white public universities may have had similar experiences during the COVID-19 pandemic.

## 6. Implications and Conclusions

The study findings show that college students were vulnerable to mental health issues during the pandemic as they feared potential exposure and illness from the COVID-19 virus and were limited in their social interactions. As predicted by the social stress theory [3], our participants’ experiences of mental health decline during the pandemic were exacerbated by social restrictions implemented by policymakers to slow down the spread of the disease. While policymakers’ focus on physical health was necessary, anti-social disease-prevention policies pose significant challenges to mental health. Mental health services were vital to some of our participants and the availability of both virtual and in-person services during the pandemic had a significant role in managing their challenges. Although all of our study participants reported having mental health concerns, not all of them used formal mental health services. We, therefore, recommend further strengthening of institution-based mental health services to proactively support student populations remotely and in-person. This could be achieved by developing messaging and tools as well as creating opportunities for individual or group therapy with the goal of maintaining social connections safely in crisis situations. We recognize that the traditional model of client-seeking-providers may not be suitable for national crisis settings and hence, mental health service providers will need to reach college students who may not have thought about seeking help. We also recommend additional institutional efforts to identify and support the most vulnerable students during crisis-induced campus closures and provide them with multiple routes to virtual and in-person mental health services.

Future research should be directed toward identifying ways of reaching college student populations for mental wellness. More research is also needed to explore trends in college student mental wellness before, during, and after the COVID-19 pandemic. College students reporting an incidence of mental health decline during the pandemic should be followed up to see whether these issues remain after the pandemic and how coping and adaptation practices evolve over time. For future outbreaks, policymakers should prioritize implementing physical and mental health interventions simultaneously, especially among college student populations. Academic institutions should prioritize setting infrastructures that would provide students with adequate mental health support for future disease outbreaks or crises. While beyond the scope of this paper, the data from this project also suggest the opportunity to explore how crises impact the building of life strategies in new circumstances especially in relation to mental health and wellness. Additionally, the findings of our paper point to the need to assess these themes presented among other vulnerable populations such as older individuals, people living with underlying conditions, and immigrants, as they are increasingly susceptible to physical and mental health issues during crises [33,34,35,36].

## Data Availability

Not applicable.

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
