# Peer review of "COVID Is Another Layer of Problematic Things”: Change, Vulnerability, and COVID-19 among University Students"

_ijerph, 2022, doi:10.3390/ijerph192315947_

Round 1

Reviewer 1 Report

In the article, an important issue is raised, the research was prepared very well, based on the extensive literature, the conclusions are justified in the research results. The article is properly prepared and therefore it is recommended for publication. However, the analysis undertaken in the article focuses on the issues discussed in popular culture. Consequently, the conclusions drawn from the research coincide with those formulated in everyday discussions. It is a pity that the authors did not focus on a specific topic, for example searching for the meaning of life in the socio-cultural context changed as a result of the pandemic, or a change in the assessment of individual values, or ways to deal with the constraints of isolation. It seems to me that it is worth adding these tips for further research perspectives. Also, when referring to limitations, it is worth noting that isolation in a pandemic prevents people from pursuing and achieving goals, while other contexts can create completely different contexts. Finally, whether the problems indicated by the respondents can be solved through therapeutic consultations? Perhaps, opportunities to build life strategies in new circumstances should be sought and offered to students?

Reviewer 2 Report

Interesting read! There will no doubt be much research yet on the effects of the Covid 19 pandemic and this paper discusses students' experiences in an easy to understand way. 

The paper could be strengthened by adding a few tables containing more quotes from a wider range of students. It would be interesting to see responses from students how had different experiences to the one already presented in the paper. 

It would also be helpful to have a table with statistical information on the participants, such as percentages for ethnic background and gender. Since the sample size was large, additional statistical analyses would also be interesting to see, such as the percentage of participants who sought professional help during the pandemic. 

As a supplementary attachment, the survey/ interview questionnaire should be included. 

Reviewer 3 Report

Thanks for inviting me to review this manuscript. A very interesting, well-organised and insightful paper. I have only one major comment:

I would like to see an additional discussion comparing your results about the vulnerability of university students and other vulnerable groups. Different population groups have been influenced by and responded to the pandemic and its interventions in very different ways (e.g., Amadasun, 2020; Purtle, 2020). For example, older people may not have sufficient access to daily necessities and activities when travel restrictions were implemented because of their low digital literacy (Liu et al., 2022a). In China, some migrant workers have been extremely vulnerable during the pandemic because they faced severe ethnic discrimination (Liu et al., 2022b). They were afraid about going to the hospital, so they concealed their Covid-like syndromes to avoid forced evictions. This did not just happen among Chinese rural-urban migrants, the concealment of Covid infection has been found in many other contexts (O’Connor & Evans, 2022). How did university students’ vulnerability during the pandemic differ from these vulnerable groups?

Reference

Amadasun, S. (2020). COVID-19 palaver: Ending rights violations of vulnerable groups in Africa. World Development, 134(1), 1-2.

Liu, Q., Liu, Z., Lin, S., & Zhao, P. (2022a). Perceived accessibility and mental health consequences of COVID-19 containment policies. Journal of Transport & Health, 101354.

Liu, Q., Liu, Z., Kang, T., Zhu, L., & Zhao, P. (2022b). Transport inequities through the lens of environmental racism: rural-urban migrants under Covid-19. Transport policy, 122, 26-38.

O’Connor, A. M., & Evans, A. D. (2022). Dishonesty during a pandemic: The concealment of COVID-19 information. Journal of Health Psychology, 27(1), 236-245.

Purtle, J. (2020). COVID-19 and mental health equity in the United States. Social psychiatry and psychiatric epidemiology, 55(8), 969-971.

Round 2

Reviewer 3 Report

I'm satisfied with the revision. Thanks for sharing.